# Quality suitability regionalization analysis of *Angelica sinensis* in Gansu, China

**Xiaoqiong Xu**[1,2], **Tiantian Zhu**[1], **Tingting Shi**[3], **Juan Chen**[4], **Ling Jin**[1,5]*

**1** School of Pharmacy, Gansu University of Chinese Medicine, Lanzhou, Gansu, China, **2** Gansu Medical College, Pingliang, Gansu, China, **3** State Key Laboratory Breeding Base of Dao-di Herbs, National Resource Center for Chinese Materia, China Academy of Chinese Medical Sciences, Beijing, China, **4** School of Pharmacy, Lanzhou University, Lanzhou, Gansu, China, **5** Research Instiute of Chinese (Tibetan)Medicinal Resources, Lanzhou, Gansu, China

* xxq198498@163.com

## Abstract

The genus *Angelica* encompasses 80 species worldwide. Among them, only *Angelica sinensis* is widely used in China and Japan. To explore the quality and geographical distribution of *A. sinensis*, we collected 1,530 plants from Gansu Province and analyzed them for their contents of chlorogenic acid (CA), ferulic acid (FA), senkyunolide I(SI), senkyunolide A (SA), senkyunolide H (SH), coniferyl ferulate (CF), ligustilide (LI), and butenyl phthalide (BP) using UPLC. We also assessed the relationship between the ecological environment and quality of *A. sinensis* through maximum entropy modeling and a geographical information system. The habitat suitability distribution demonstrated that the most influential ecological factors for the growth of *A. sinensis* were altitude, precipitation during March, May, and December, precipitation during the wettest month, and the soil pH. The most suitable areas for cultivation are concentrated to the south of Gansu Province, including Linxia Hui Autonomous Prefecture, Dingxi City, Tianshui City, south of Wuwei City, east of Gannan Tibetan Autonomous Prefecture, north of Longnan City, and northwest of Pingliang City. The quality suitability regionalization analysis divulged that the most influential ecological factors for the index components of *A. sinensis* were the altitude, sunshine, rainfall, temperature, and soil pH. The highest quality *A. sinensis* grow in Dingxi City, Tangchang, Lixian, and Wen counties in Longnan City, Wushan County in Tianshui City, Lintan, Zhouqu, and Zhuoni counties in Gannan Tibetan Autonomous Prefecture, Kangle and Linxia counties in Linxia Hui Autonomous Prefecture. The experiment yielded highly accurate results (accuracy of 0.955), suggesting that the results were consistent with the actual distribution of *A. sinensis* in Gansu. The inferences of this research will naturally draw the attention of the authorities in the fields of natural resources and agriculture departments and provide a scientific basis for the rational selection of *A. sinensis* cultivation areas.

**Data Availability Statement:** All relevant data are within the paper and its Supporting information files.

**Funding:** Initials of the author who received each award: L J Grant number awarded to the author:

81360615 The full name of each funder: National Natural Science Foundation of China URL of each funder website: http://www.nsfc.gov.cn/ Specific grant numbers: ¥520000.

**Competing interests:** The authors have declared that no competing interests exist.

## Introduction

The root of *Angelica sinensis* (Oliv.) Diels (Umbelliferae family) is used as a health supplement and drug in Asian countries and a dietary supplement in women's care in Europe [1–3]. *A. sinensis* has been cultivated in China for more than 2,000 years. The root of this herb is an important traditional Chinese medicine, primarily prescribed for tonifying the blood and treating anemia, rheumatism, and menstrual disorders. This plant is primarily cultivated in the Gansu, Yunnan, Sichuan, Shaanxi, and Hubei Provinces in China [4–6]. Owing to its high commercial value and large export market, wild *A. sinensis* has been overharvested. Currently, the cultivated varieties of herbs are preferably used for medicinal purposes. *A. sinensis* is a typical habitation-dominated medicinal herb. The soil pH, rainfall, temperature, altitude, and other ecological factors greatly influence the active components of *A. sinensis* [7–9]. For economic benefits, farmers have blindly expanded the cultivation areas of *A. sinensis*, even in places where the environment is not suitable for the growth of *A. sinensis*, which has eventually deteriorated the quality of *A. sinensis*. Therefore, the relationship between the quality and ecological environment should be established to obtain quality suitability map. In this study, Maxent and ArcGIS techniques have been used for the first time to conduct a regionalization study on the quality of *A. sinensis*. Using Maxent modeling, we identified the suitable areas in Gansu for cultivating high-quality *A. sinensis* species and promoting their cultivation in such suitable areas.

## Materials and methods

In this article, 1,530 sites were selected for collecting samples. Sampling point informations are listed in S1 Appendix. The contents of CA, FA, SI, SA, SH, CF, LI, and BP were determined. Using MaxEnt, we calculated the growth suitability of *A. sinensis* and identified suitable cultivation areas. To correlate the quality of the medicinal plants with their growth environment, we analyzed the positive and negative effects of the habitat conditions on plant growth based on the 8 index component accumulation process and prepared a map of quality suitability of *A. sinensis* in Gansu.

### Survey areas and species occurrence records

The survey areas are in Gansu Province, China, at the confluence of the Qinling Mountains, Loess Plateau, and Qinghai-Tibet Plateau with abundant vegetation. The altitude varies between 1,800 and 3,247 m, which lies between 33˚58′ N to 38˚40′ N latitude and 102˚57′ E to 104˚30′ E longitude. The survey areas receive an annual average of 36.6–734.9 mm precipitation at an average temperature of 0–15˚C. The sub-tropical monsoon season of the study area extends from June to August, followed by a dry season from November to March.

Based on the preliminary study and planting scale of *A. sinensis* in Gansu, 1,530 cultivated samples of *A. sinensis* were collected from Dingxi City, Gannan Tibetan Autonomous Prefecture, Longnan Prefecture, Linxia Hui Autonomous Prefecture, Wuwei City, Tianshui City, Zhangye City, and Lanzhou City between October and November 2018. During the collection process, the latitude, longitude, and habitat information were recorded using a handheld GPS device, and the information on sampling points was stored in the table. All the samples were crushed after drying, sifted, cryopreserved (2–8˚C), and later used to determine the 8 index components. According to the principle of uniform representation, the sampling points were spaced approximately 500 to 800 meters apart. Because the index composition changes with the growth duration, the samples are all 3-year-olds, so that the variables are controlled within a reasonable range. Species occurrence records were displayed in Fig 1.

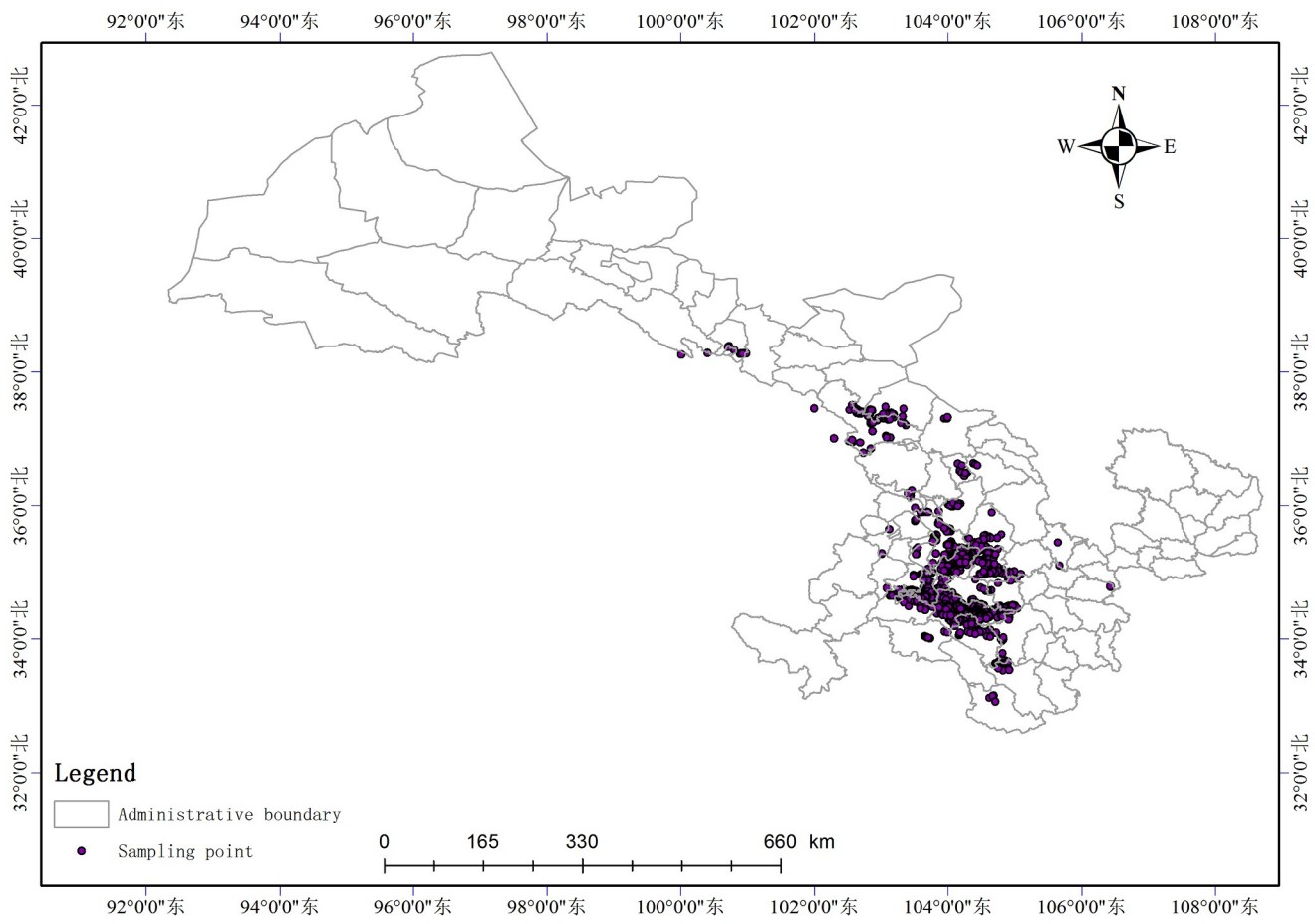

**Fig 1. Species occurrence records.**

### Acquisition and selection of ecological factors

The data of 60 ecological factors, including meteorological data [10], soil type [11], topographical data [12], and vegetation type [13], were collected. A total of 55 continuous variables and 5 categorical variables were analyzed (Table 1). The ecological factor database used in this study has been derived from the "Chinese Medicine Resource Spatial Information Grid Database". The map has been downloaded from the department of natural resources of Gansu Province (http://zrzy.gansu.gov.cn/) (resolution-1:650), Map review number: Gan S(2017)64. The attributes for each sampling point were calculated before analysis using correlation software. Correlation coefficients were calculated among 60 ecological factors, and the ecological factors that contained correlation coefficients that were less than 0.8 were retained. Nine ecological factors fulfilled these requirements.

### Operation and accuracy of testing by MaxEnt

The sampling information of *A. sinensis* and 60 ecological factors were added to MaxEnt. A total of 25% of the distribution data were randomly selected as the test set and the rest as the training set [14]. The maximum number of iterations was set as $10^6$, and the convergence threshold was set at 0.00005, while the other parameters were set as the default. These calculations were repeated five times. The contribution of each ecological factor to the growth of *A.*

**Table 1. List of the environmental variables used to develop the model.**

| Code | Environmental variables | Unit |
|---|---|---|
| bio1 | Annual Mean Temperature | ˚C |
| bio2 | Mean Diurnal Range (Mean of monthly (max temp—min temp)) | ˚C |
| bio3 | Isothermality (bio 2/bio 7) (* 100) % | ˚C |
| bio4 | Temperature Seasonality (standard deviation *100) | ˚C |
| bio5 | Max Temperature of Warmest Month | ˚C |
| bio6 | Min Temperature of Coldest Month | ˚C |
| bio7 | Temperature Annual Range (bio5-bio6) | ˚C |
| bio8 | Mean Temperature of Wettest Quarter | ˚C |
| bio9 | Mean Temperature of Driest Quarter | ˚C |
| bio10 | Mean Temperature of Warmest Quarter | ˚C |
| bio11 | Mean Temperature of Coldest Quarter | mm |
| bio12 | Annual Precipitation | mm |
| bio13 | Precipitation of Wettest Month | mm |
| bio14 | Precipitation of Driest Month | mm |
| bio15 | Precipitation Seasonality (Coefficient of Variation) | 1 |
| bio16 | Precipitation of Wettest Quarter mm | mm |
| rz | Annual sunshine duration | h |
| altitude | Elevation | m |
| index-hi | Humidity index | mm/˚C |
| index-ci | Cold index | ˚C |
| index-wi | Warm index | ˚C |
| tmean | Mean Temperature of Each Month | ˚C |
| prec | Precipitation of Each Month | mm |
| ntl | Amount of clay | % |
| yjthl | Soil organic carbon content | % |
| pH | Soil pH | 1 |

*sinensis* was determined. The weight of each climate factor was verified by the Jackknife procedures in MaxEnt. An ROC analysis can evaluate the utility of a model. The area under the curve (AUC) is an effective threshold-independent index to discriminate the presence from absence (or background) [15]. The accuracy of the results is proportional to the AUC value. AUC≥0.9 indicates an excellent performance by the model [16]. The results of the simulation output from MaxEnt software ranged from 0 to 1, the closer was the value to 1, the greater was the probability of the existence of the species.

## Habitat suitability distribution

The regionalization model demonstrated the distribution of suitable habitat for *A. sinensis*. We collected the habitat suitability values from 1,530 distribution points. The habitat suitability was then divided into three levels following the Natural Breaks (Jenks) method by marking in three colors for inappropriate areas (0–0.143), appropriate areas (0.144–0.378), and optimum areas (0.379–0.530). The layers with ecological overlays were loaded in ArcMap. The attributes of the setting layer in the symbol system are set based on the classification of habitat suitability. The distribution map of *A. sinensis* in Gansu Province was extracted by spatial analysis technology in ArcGIS.

## Instruments, reagents and reference substances

ACQUITY UPLC (Waters, Milford, MA, USA), CP114 Electronic analytical balance (Ohaus Instrument Co. LTD, Shanghai, China), Type 101 electric blast drying oven (Beijing Yong-guang Medical Equipment Factory, Beijng, China), 0.22 m microporous filter membrane (Shanghai Amspectral Experimental Technology Co., LTD, Shanghai). The reference samples of chlorogenic acid (CAS: 327-97-9), ferulic acid (CAS: 1135-24-6), ligustilide (CAS: 4431-01-0), senkyunolide I (CAS:94596-28-8), senkyunolide A (CAS: 62006-39-7), senkyunolide H (CAS: 94596-27-7), coniferyl ferulate (CAS: 63644-62-2), and butenyl phthalide (CAS: 551-08-6) were purchased from Chengdu Reffens Biotechnology Co., Ltd (Chengdu, China). The purity of the reference substances was greater than 98%. The methanol was chromatographic grade; the water was ultrapure, and the other reagents were analytically pure.

## Sample preparation

*A. sinensis* powder (5.0 g) was accurately weighed and placed in a conical flask with a stopper. The sample was extracted with 50% methanol (50 mL) in an ultrasonic bath for 45 min. After refilling the volume, the extract was filtered through filter paper and a 0.22 mm filter membrane for analysis.

## Chromatographic conditions

UPLC-PDA analysis was performed on an ACQUITY UPLC system, PDA detector. An ACQUITY UPLC $^®$BEH $C_{18}$ column (2.1 mm×50 mm, 1.7 μm, Waters) was applied for the separation. The mobile phase, composed of A (methanol) and B (acetic acid/water, 1/99, v/v), was run following a gradient elution: (0–5 min, 5%→30% A; 5–6.5 min, 30%→35% A; 6.5–8.5 min, 35%→50%A; 8.5–14 min, 50%→80%A; 14–17 min, 80% A;17~22 min, 80%→100%A; and 22–27 min, 100%→5%A). The column temperature was set at 30˚C. The flow rate was set at 0.3 mL/min. The injection volume was 3 μL, and the detection wavelength was 270 nm.

## The relationship model

A stepwise regression analysis was conducted between the ecological factor data and the index components to establish a prediction model for CA, FA, SI, SA, SH, CF, LI, and BP. The contents of 8 index components were used as the dependent variables and the ecological factors as independent variables to fit a linear relationship and set up the model for quality suitability regionalization for *A. sinensis*.

## Comprehensive quality evaluation

Based on the distribution of habitat suitability of *A. sinensis* in Gansu Province, the unsuitable distribution areas were removed, and the spatial distribution maps of the index components in the suitable areas were obtained using ArcGIS. According to the regulations specified in the current Chinese Pharmacopoeia (2015), organic acids and volatile oils (the main effective components in *A. sinensis*) were used as the index ingredients to determine the contents in the *A. sinensis*, CA, FA, SI, SA, SH, CF, LI, and BP are representative of these compounds. The content of ferulic acid in *A. sinensis* should not be less than 0.05%, and the content of essential oils should not be less than 0.4% (ml/g). We identified the ecological factors that affected the index content and investigated the effects of the ecological factors on the accumulation of the organic acids and volatile oils. Based on the ArcGIS fuzzy superposition function, the spatial distribution superposition map of all the index components was obtained, and the quality of *A. sinensis* was comprehensively evaluated.

**Table 2. The accumulated contribution of each ecological factor.**

| Environmental variables | Percentages of contribution (%) | Accumulating percentages of contribution (%) | Environmental variables | Percentages of contribution (%) | Accumulating percentages of contribution (%) |
|---|---|---|---|---|---|
| altitude | 41.9 | 41.9 | bio3 | 1.1 | 95.8 |
| prec3 | 15.9 | 57.8 | tmean11 | 0.8 | 96.6 |
| prec5 | 13.9 | 71.7 | bio4 | 0.8 | 97.4 |
| bio13 | 8.1 | 79.8 | Prec6 | 0.8 | 98.2 |
| prec12 | 6.1 | 85.9 | tmean12 | 0.8 | 99.0 |
| pH | 3.8 | 89.7 | prec10 | 0.8 | 99.8 |
| bio2 | 2.6 | 92.3 | tmean9 | 0.2 | 100 |
| bio7 | 2.4 | 94.7 | — | — | — |

## Results

### The key environmental factors and modeling results

The key environmental factors were determined according to the contributions to the modeling process using the jackknife test (Table 2). The following nine environmental variables with the cumulative contributions of 95.8% were screened as the key environmental factors: altitude (41.9%), precipitation during March, May, and December (15.9%, 13.9%, and 6.1%), precipitation during the wettest month (8.1%), pH (3.8%); mean diurnal range (2.6%), temperature annual range (2.4%), and isothermality (1.1%). To eliminate the influence of collinearity on the modeling process and interpretation of results, a Pearson correlation analysis was conducted for the nine environmental factors. The result demonstrated that the Pearson correlation coefficients for the eight environmental factors were all less than 0.8 (Table 3). Thus, the nine environmental variables described above were selected as the key environmental factors to reconstruct the MaxEnt model. The AUC value of the ROC curve reaches 0.957, indicating an excellent prediction accuracy of the model. This model can reliably define suitable areas for the cultivation of *A. sinensis* in Gansu Province, China.

### Habitat suitability distribution

The habitat suitability distribution of *Angelica sinensis* (Fig 2) was plotted after classifying the growth suitability of *A. sinensis* at different levels. The optimal areas for growth of *A. sinensis* were located to the south of Wuwei City, Linxia Hui Autonomous Prefecture, Dingxi City, east

**Table 3. Pairwise Pearson's correlation coefficients of the environmental variables.**

| code | altitude | bio2 | bio3 | bio7 | bio13 | prec3 | prec5 | prec12 |
|---|---|---|---|---|---|---|---|---|
| bio2 | .358** | — | — | — | — | — | — | — |
| bio3 | .585** | .776** | — | — | — | — | — | — |
| bio7 | -.182** | .615** | -0.007* | — | — | — | — | — |
| bio13 | .442** | -.096** | .472** | -.766** | — | — | — | — |
| prec3 | .302** | -.244** | .374** | -.870** | .734** | — | — | — |
| prec5 | .363** | 0.004 | .584** | -.747** | .896** | .717** | — | — |
| prec12 | .131** | -.553** | -.126** | -.730** | .655** | .704** | .468** | — |
| pH | -.301** | .057* | -.077** | .198** | -.199** | -.194** | -.180** | -.200** |

$^*$ $P < .05$.

$^{**}$ $P < .01$.

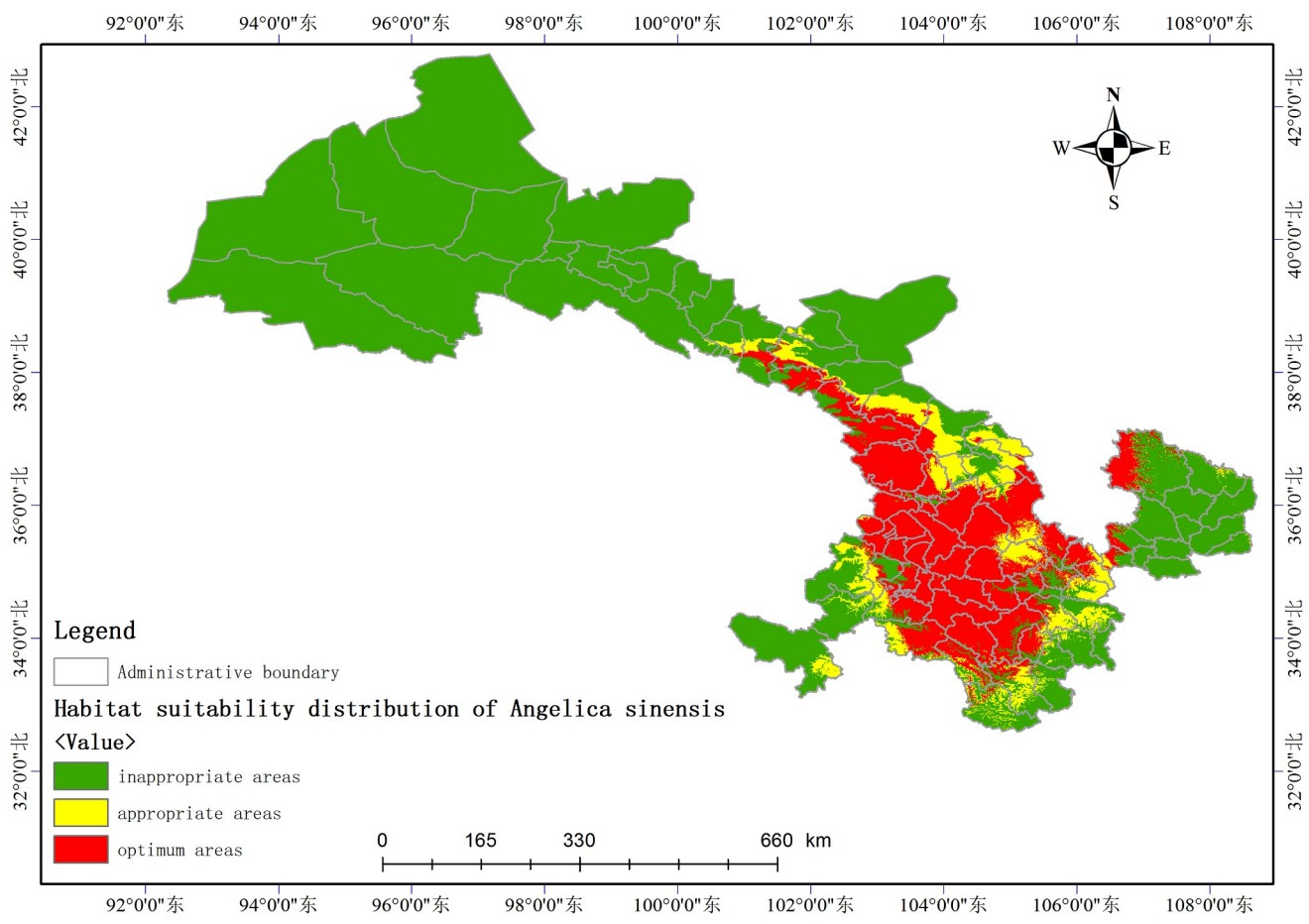

**Fig 2. Habitat suitability distribution of *Angelica sinensis*.**

of Gannan Tibetan Autonomous Prefecture, north of Longnan City, surrounding Tianshui City, and northwest of Pingliang City. The appropriate areas for the growth of *A. sinensis* include Lanzhou City and southeast of Wuwei City. The remaining areas were predicted to be inappropriate for the growth of *A. sinensis*. In Fig 1, the purple dots represent the actual presence of *A. sinensis*. The predicted suitable distribution range was determined based on the actual presence of plants. The response curves represent the relationship between environmental variables and suitability of habitats, they help us understand the ecological niche of a species. The ranges of suitability for environmental variables were identified by the threshold of normally suitable habitats. Response curves of the important ecological factors are illustrated in (S1 File), and the suitable range for each variable is shown in (Table 4). *A. sinensis* grew at the altitude from 2,000 to 3,000 m, with an optimal altitude of 2760 m, and the precipitation during March was recorded from 5 to 20 mm with optimal precipitation of 16 mm. In contrast, the suitable range of isothermality was from 32 to 38% with an optimal value of 35%.

## Results of the determination of content

The contents of CA, FA, SI, SA, SH, CF, LI, and BP in *A. sinensis* samples agreed with the regulations specified in the current Chinese Pharmacopoeia (2015), and they can be used as

Table 4. The suitable range and optimum value of each environmental variable.

| Environmental variables | Suitable range | Optimal value |
|---|---|---|
| altitude | 2000–3000 m | 2760 m |
| prec3 | 5–20 mm | 16 mm |
| prec5 | 50–100 mm | 80 mm |
| bio13 | 80–120 mm | 120 mm |
| prec12 | 1–5 mm | 2 mm |
| pH | 6.5–8 | 6.3 |
| bio2 | 10–12˚C | 11˚C |
| bio7 | 32–38˚C | 35˚C |
| bio3 | 32–38% | 35% |

indicators for the quality suitability regionalization of *A. sinensis*. The UPLC chromatograms of the mixed standard and samples are shown in S2 File.

## The relationship models of eight index components

The eight regression equations and the regression coefficient of each factor are all significant (Table 5). The eight index components content can be predicted by the relationship models.

## Comprehensive quality evaluation of *A. sinensis*

Based on the relationship model between the index components and ecological factors of *A. sinensis*, using the spatial analysis function of ArcGIS v.10.5, the distribution of 8 index components has been estimated and depicted in (Figs 3–10). These figures show that in a suitable area, the contents of FA, SI, SH decrease from south to north, the content of CA decreases from east to west, the content of LI decreases from west to east. From north to south, the content of SA decreases, the distribution regularity of BP is not strong, and the content of CF in the suitable area is more consistent. Quality suitability regionalization of *Angelica sinensis* are

Table 5. The relationship models of eight index components.

| NO. | Regression equation | F test result of the regression equation | T test results of the regression coefficients |
|---|---|---|---|
| 1 | $y_{FA} = 0.184 + 0.001^*prec12 − 0.009^*\ pH$ | $P = 5.10\ e\ \text{-}8 < 0.05$ | 8.75e-9, 0.032<br>all less than 0.05 |
| 2 | $y_{CA} = \text{-}1.308 + 0.011^*index\text{-}wi + 0.008^*bio2 + 0.0291^*index\text{-}hi$ | $P = 1.39\ e\ \text{-}6 < 0.05$ | 1.37e-5, 0.003, 0.025<br>all less than 0.05 |
| 3 | $y_{SI} = 0.001^*alttitude − 0.003^*\ pH + 0.005^*bio2 + 0.001^*prec3$ | $P = 8.88\ e\ \text{-}13 < 0.05$ | 0.005, 0.014, 0.006, 0.019<br>all less than 0.05 |
| 4 | $y_{SH} = \text{-}0.124 + 0.002^*prec3 − 0.002^*rz + 0.002^*altitude + 0.003^*prec5 − 0.002^*bio6 − 0.002^*bio2$ | $P = 1.06\ e\ \text{-}17 < 0.05$ | 0.011, 2.56e-10, 4.22e-14<br>0.011, 1.63e-13, 4.38e-13<br>all less than 0.05 |
| 5 | $y_{SA} = 0.002 + 1.345e\text{-}5^*prec12$ | $P = 0.032 < 0.05$ | 0.032 < 0.05 |
| 6 | $y_{CF} = 0.076 − 2.204e\text{-}5^*rz$ | $P = 0.043 < 0.05$ | 0.043 < 0.05 |
| 7 | $y_{LI} = 0.471 + 0.02^*tmean7$ | $P = 0.028 < 0.05$ | 0.028 < 0.05 |
| 8 | $y_{BP} = 0.017^*altitude − 0.025^*rz + 0.022^*prec4$ | $P = 1.66\ e\ \text{-}24 < 0.05$ | 0.003, 3.47e-15, 1.09e-8<br>all less than 0.05 |

Note: $y_{FA.}, y_{CA}, y_{SI}, y_{SH}, y_{SA}, y_{CF}, y_{LI}, y_{BP}$ denotes the contents of chlorogenic acid, ferulic acid, senkyunolide I, senkyunolide H, senkyunolide A, coniferyl ferulate, ligustilide, and butenyl phthalide.

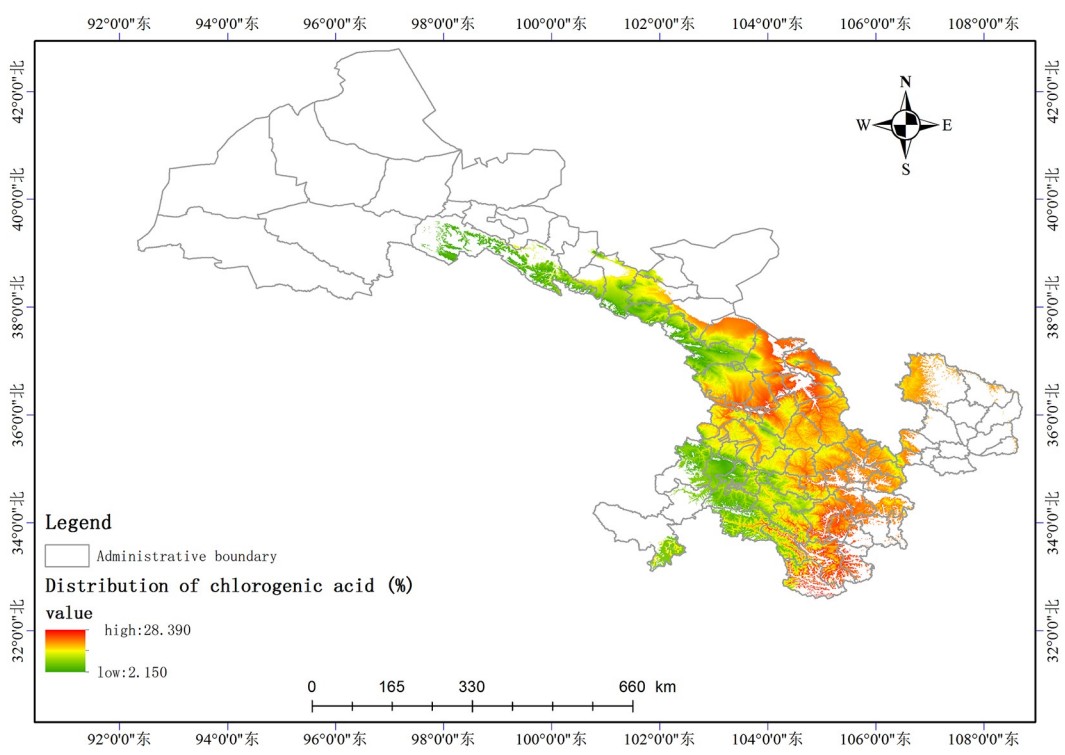

**Fig 3. Distribution of chlorogenic acid (%).**

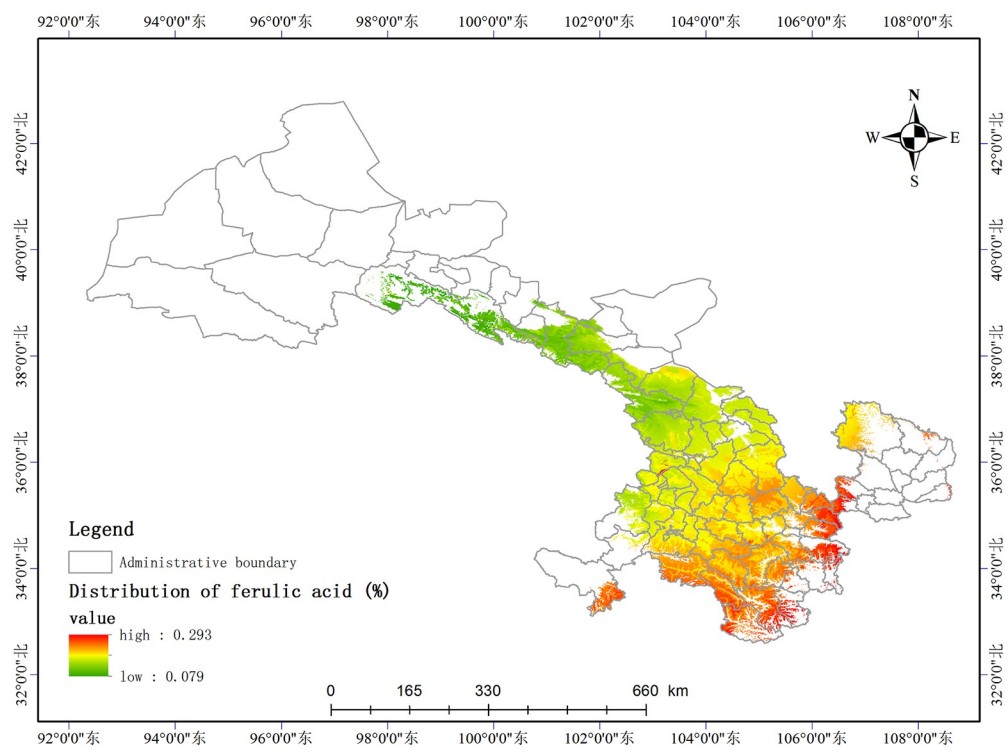

**Fig 4. Distribution of ferulic acid (%).**

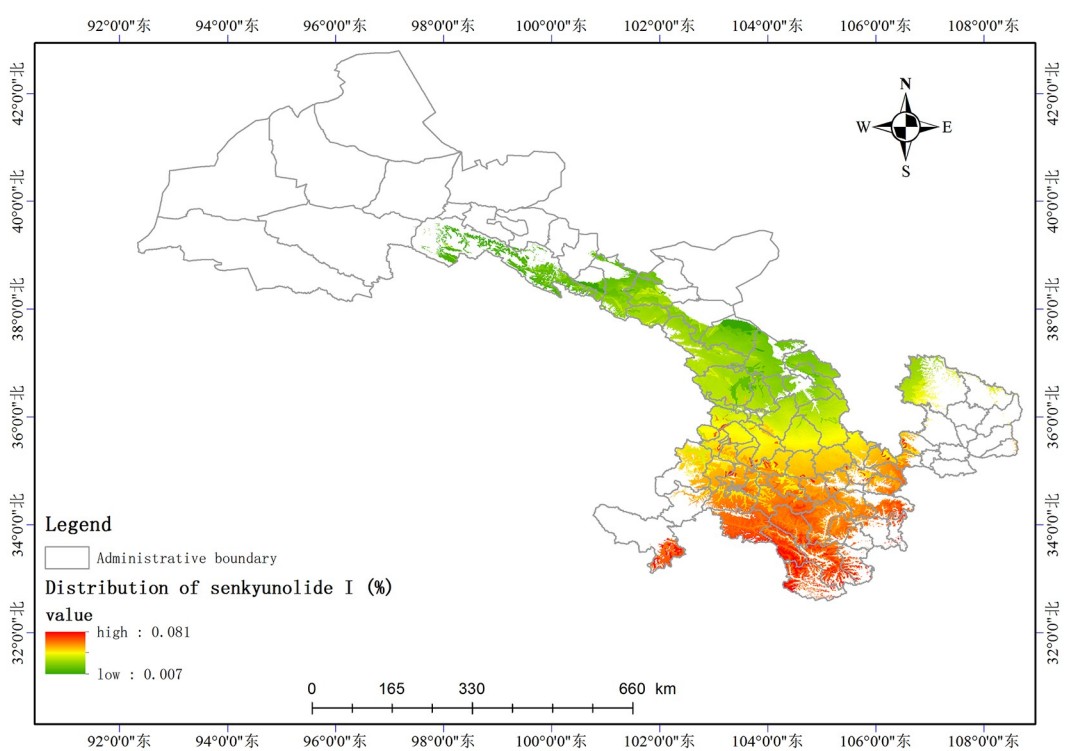

**Fig 5. Distribution of senkyunolide I (%).**

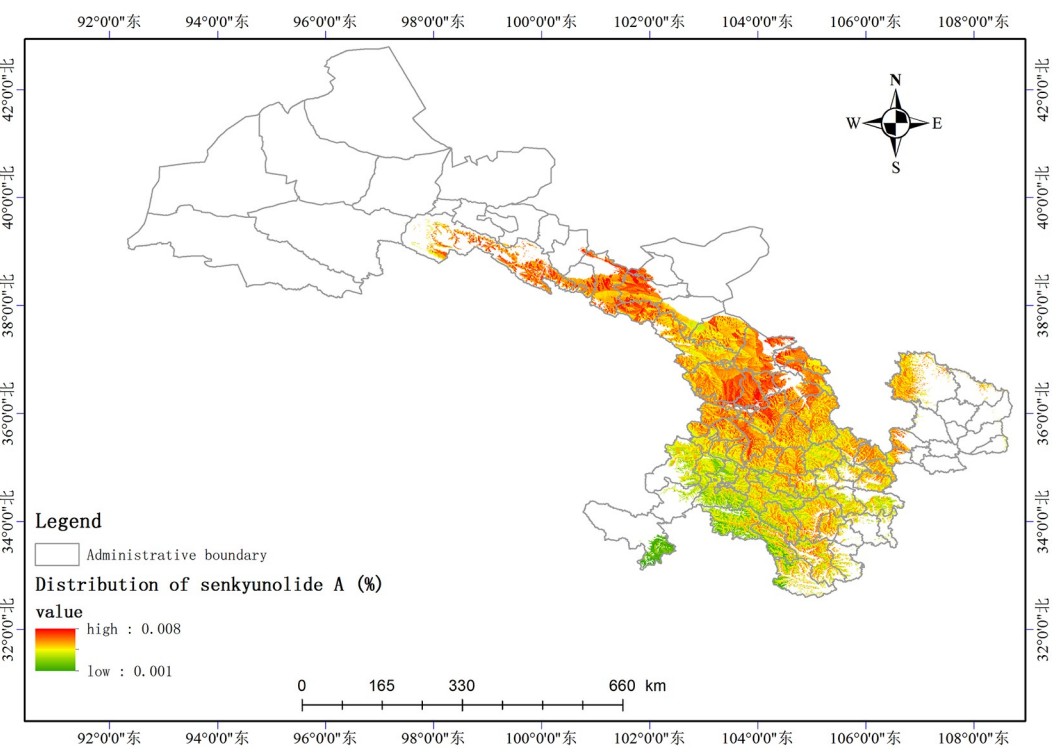

**Fig 6. Distribution of senkyunolide A (%).**

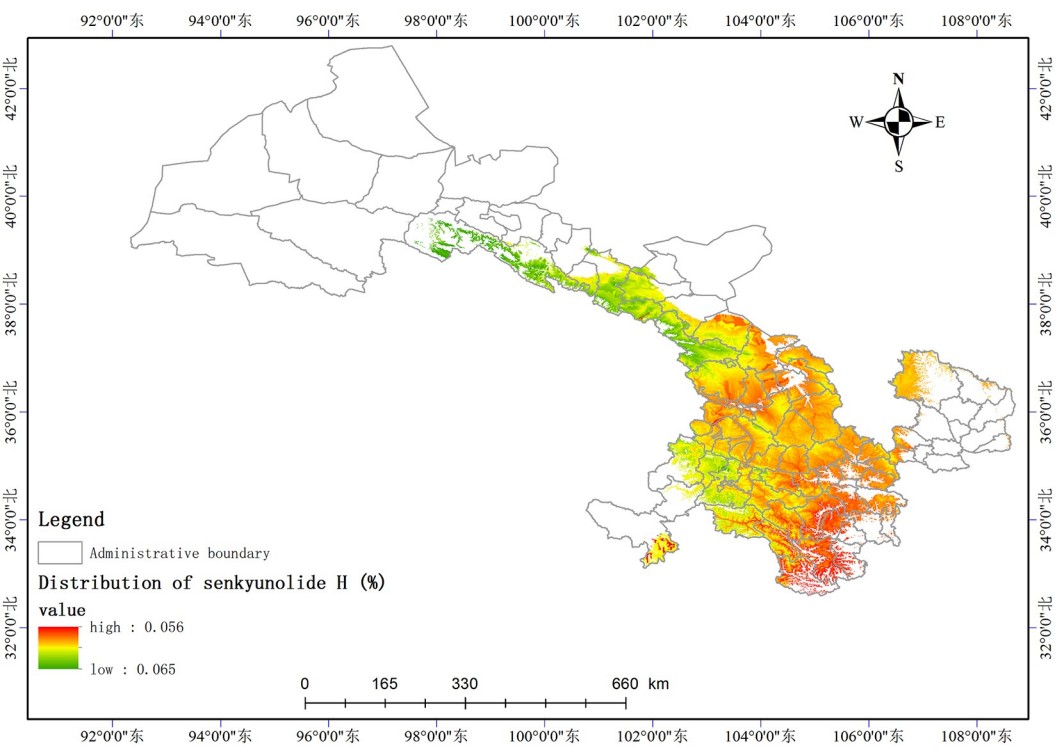

**Fig 7. Distribution of senkyunolide H (%).**

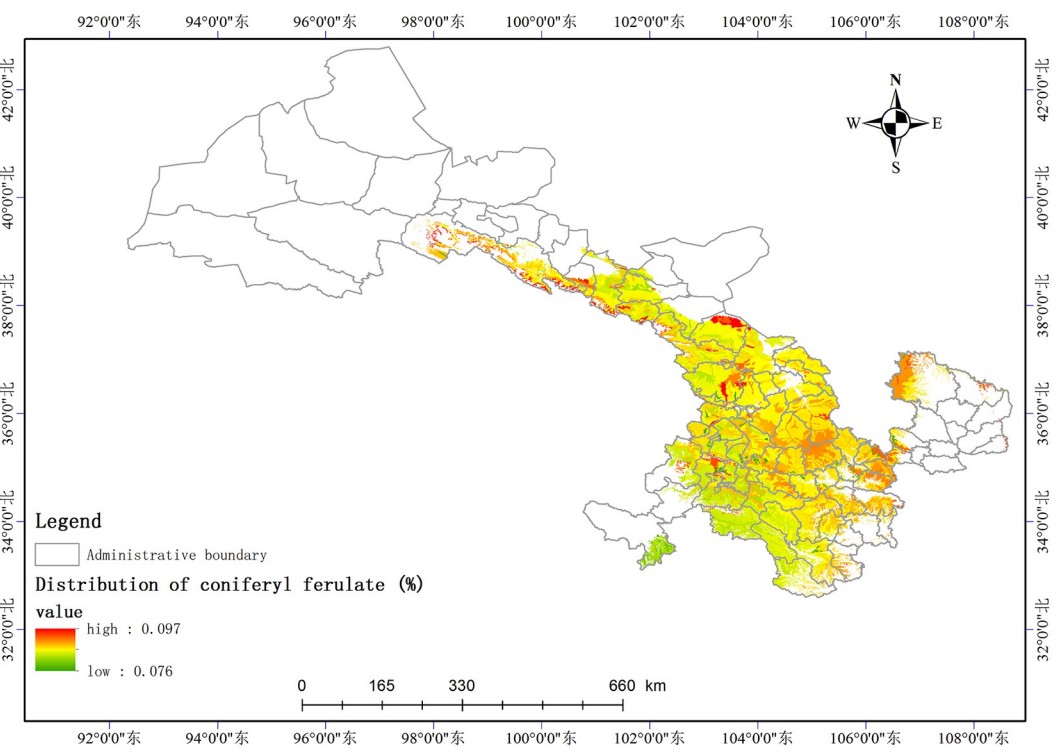

**Fig 8. Distribution of coniferyl ferulate (%).**

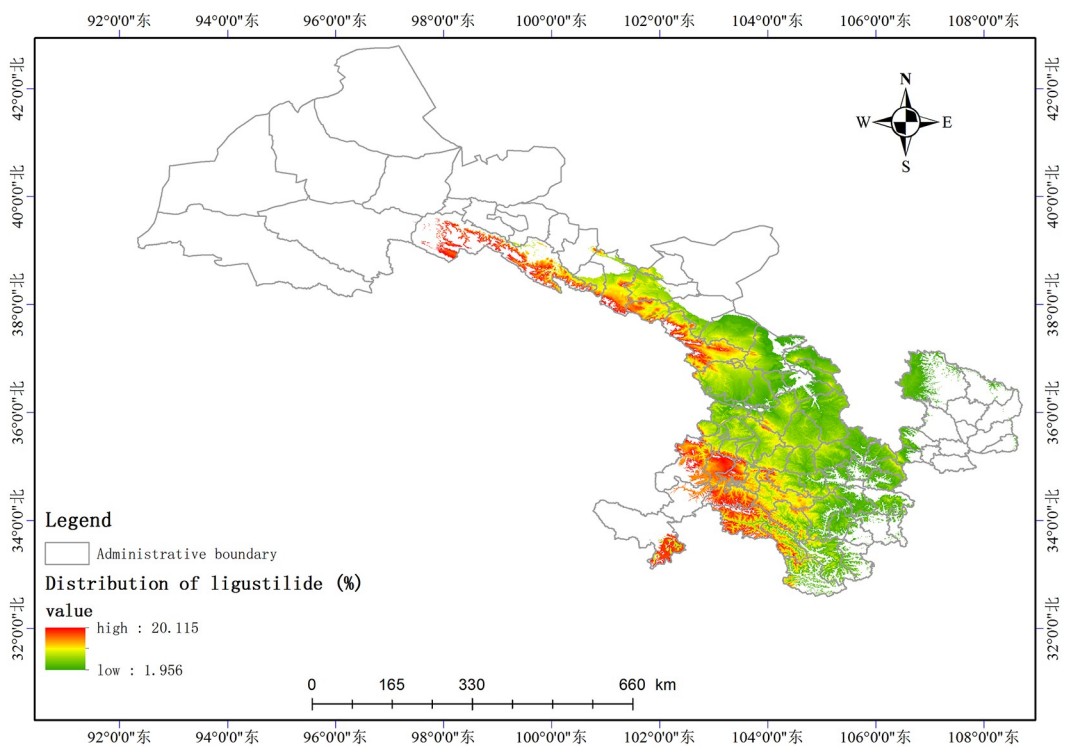

**Fig 9. Distribution of coniferyl ferulate (%).**

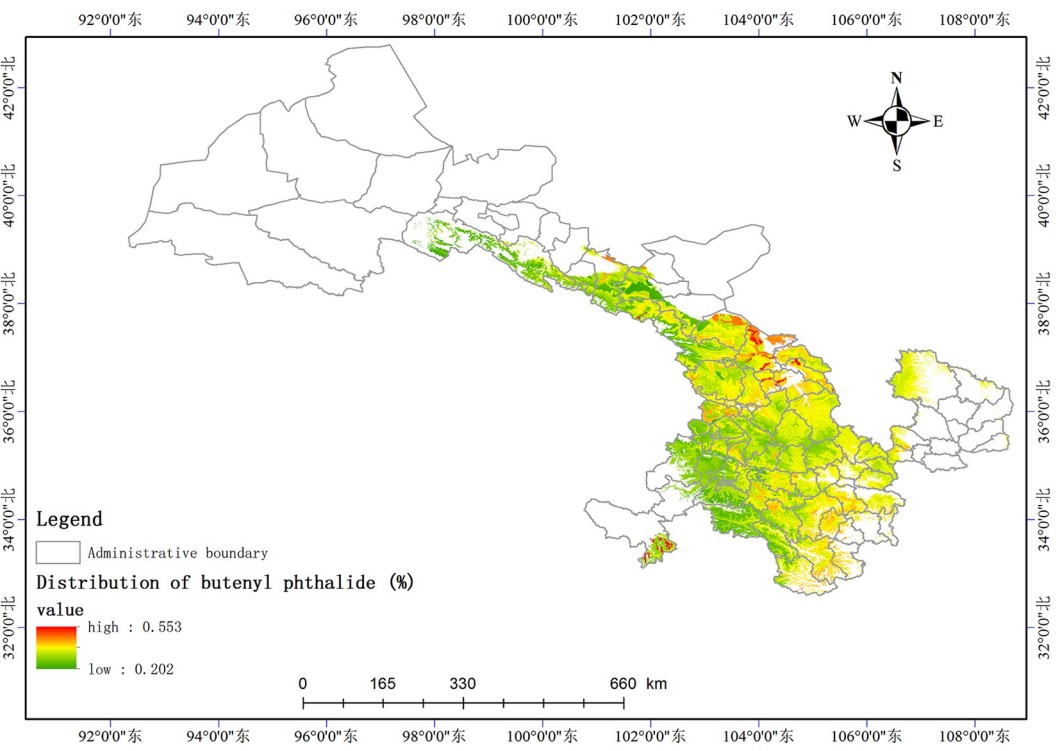

**Fig 10. Distribution of butenyl phthalide (%).**

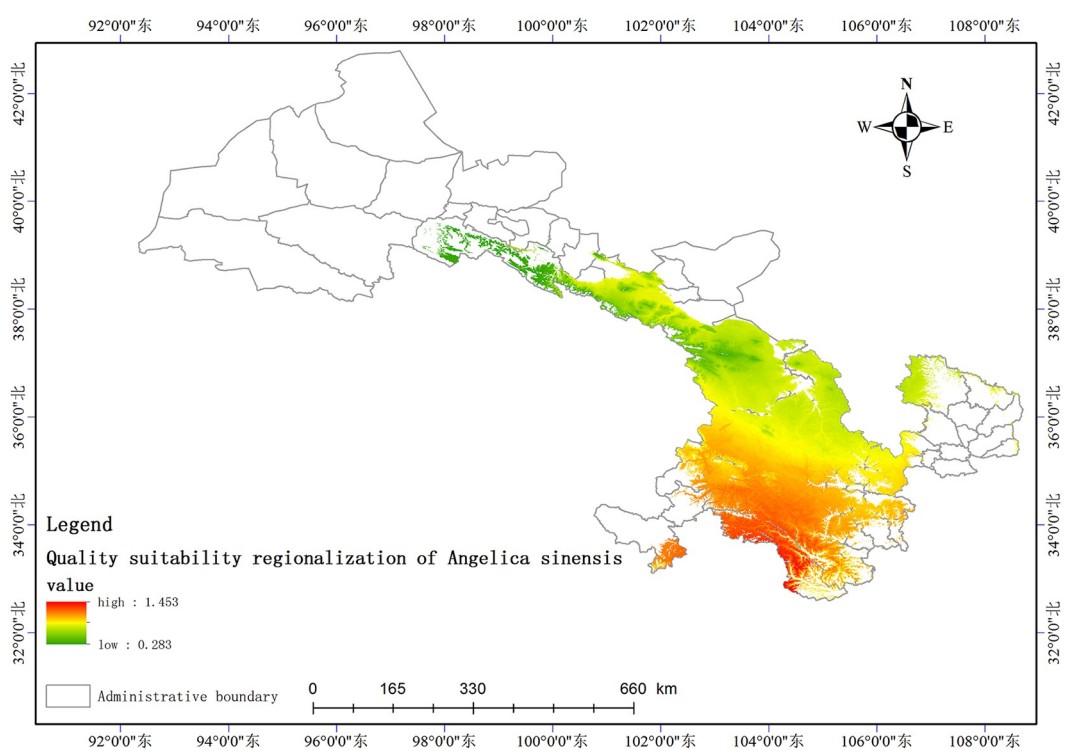

**Fig 11. Quality suitability regionalization of *Angelica sinensis*.**

shown in (Fig 11). Fig 11 indicates that *A. sinensis* species cultivated in Dingxi city, south of Tianshui, west of Longnan, east of Wuwei and Gannan Tibetan Autonomous Prefecture, and Linxia Hui Autonomous Prefecture contain the highest levels of active constituents.

## Discussion

### Effects of the meteorological factors

According to the relationship model of eight index components, the content of FA positively correlated with prec12 but negatively correlated with the soil pH. The content of CA positively correlated with index-wi, bio2, and index-hi. The content of SI positively correlated with altitude, bio2, and prec3 but negatively correlated with the soil pH. The content of SH positively correlated with prec3, prec5, and altitude but negatively correlated with rz, bio6, and bio2. The content of SA positively correlated with prec12. The content of CF negatively correlated with rz. The content of LI positively correlated with tmean7. The content of BP positively correlated with prec4 and altitude but negatively correlated with rz. The influence of temperature and moisture on the content of index components of *A. sinensis* is highly significant [17]. A moderately low temperature aids in the accumulation of organic acids [18], and longer periods of sunshine did not increase the volatile oil composition [19]. This may be related to the fact that *A. sinensis* is a low temperature, short sunshine plant [20]. The total weight of precipitation in all the ecological factors is 43.93%, Gansu is located in the arid area of northwest China with an annual precipitation of 36.6–734.9 mm, which decreases from southeast to northwest. The precipitation is concentrated in the summer, comprising 50–70% of the annual precipitation. Thus, the moderate precipitation in spring and winter positively correlated with the index composition.

## Effects of topographical and soil factors

The results of this study showed that the topographic and soil factors greatly affected the content of the index components. The weight of the altitude on *A. sinensis* distribution reached 41.88%. The soil pH was 3.8%. All the index components positively correlated with the altitude and negatively correlated with the soil pH. The altitude was within 2,000–3,000 m; the higher the altitude, the greater the accumulated content of the index components. As the altitude increases, photosynthetic products are distributed earlier into the roots, and the rate of accumulation of dry matter increases [21]. The appropriate soil pH of *A. sinensis* is 6.5–8. Investigation found that the soil in the Gansu production area was alkaline; thus, the content of the index components negatively correlated with the soil pH [22].

## Comparison of the study results

The map of the quality suitability regionalization analysis shows that high contents of the index compounds are present in *A. sinensis* that grow in Minxian, Zhangxian, Weiyuan, Lintao, Dangchang, Lixian, Wenxian, Wushan, Lintan, Zhouqu, Zhuoni, Kangle, Linxia, Jishi Mountain, Tianzhu, and Gulang. A comparison between the areas, selected based on the suitability of regionalization and the traditional farming areas of *A. sinensis*, suggests that most of the areas with high suitability are those in the traditional growing areas [23]. Although the range of sampling was expanded to maximize the sampled specimens of *A. sinensis*, some wild *A. sinensis* specimens from unknown areas were not included in this study owing to some limitations, such as ecological conditions. These unknown areas will be included in our future studies.

## Conclusion

The suitable areas for *A. sinensis* were predicted successfully by the MaxEnt model and ArcGIS. *A. sinensis* is primarily distributed to the south of Gansu Province, and the distribution areas are concentrated. The high suitability regions are located primarily to the south of Wuwei City, Linxia Hui Autonomous Prefecture, Dingxi City, east of Gannan Tibetan Autonomous Prefecture, north of Longnan City, surrounding Tianshui City, northwest of Pingliang City, and the sub-suitable areas include Lanzhou City, southeast of Wuwei City. The distribution of *A. sinensis* was affected by some key environmental variables, including the altitude, precipitation during March, May, December, mean diurnal range, isothermality, annual temperature range, precipitation during the wettest month, and soil pH. The duration of annual sunshine affects the content of ferulic acid in *A. sinensis*. Longer periods of sunshine are unfavorable for the biosynthesis of ferulic acid. The volatile oil content of *A. sinensis* is greatly affected by the soil pH, altitude, precipitation in March, April, May, and December, annual sunshine duration, and temperature. Quality suitability regionalization of *A. sinensis* in Gansu Province shows that *A. sinensis* specimens from the south of Gansu, such as Dingxi City, Tangchang County, Lixian County, the Wen county in Longnan City, Wushan county in Tianshui City; the Lintan, Zhouqu, Zhuoni county in Gannan Tibetan Autonomous Prefecture; Kangle and Linxia county in Linxia Hui Autonomous Prefecture, contain the best overall quality and the highest levels of active constituents.

## Supporting information

**S1 Appendix. Sampling point informations.**
(PDF)

**S1 File. Response curves of the important ecological factors.**
(PDF)

**S2 File. UPLC chromatograms of the mixed standard (A) and _Angelica sinensis_ (B).** Note: 1. Chlorogenic acid, 2. Ferulic acid, 3. Senkyunolide I, 4. Senkyunolide H, 5. Senkyunolide A, 6. Coniferyl ferulate, 7. Ligustilide, 8. Butenyl phthalide.
(PDF)

## Acknowledgments

We thank Rui Huang for reviewing the draft and polishing language of this manuscript and Xiaobo Zhang and Shaoyang Xi for their helpful suggestions on the research methodologies.

## Author Contributions

**Data curation:** Xiaoqiong Xu, Juan Chen.

**Formal analysis:** Xiaoqiong Xu.

**Funding acquisition:** Ling Jin.

**Investigation:** Xiaoqiong Xu, Tiantian Zhu.

**Methodology:** Xiaoqiong Xu.

**Project administration:** Ling Jin.

**Writing – original draft:** Xiaoqiong Xu, Tingting Shi.

**Writing – review & editing:** Xiaoqiong Xu.

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
