## [Decision Letter · Decision Letter 0]

22 Oct 2020

PONE-D-20-27049

Quality suitability regionalization analysis of Angelica sinensis in Gansu , China

PLOS ONE

Dear Dr. XU,

Thank you for submitting your manuscript to PLOS ONE. After careful consideration, we feel that it has merit but does not fully meet PLOS ONE’s publication criteria as it currently stands. Therefore, we invite you to submit a revised version of the manuscript that addresses the points raised during the review process.

We look forward to receiving your revised manuscript.

Kind regards,

Vassilis G. Aschonitis

Academic Editor

PLOS ONE

Journal Requirements:

"5172".

4.We note that [Figure(s) 2, 4, 7, 8, 9, 10, 11, 12, 13, 14, and 15] in your submission contain [map/satellite] images which may be copyrighted. All PLOS content is published under the Creative Commons Attribution License (CC BY 4.0), which means that the manuscript, images, and Supporting Information files will be freely available online, and any third party is permitted to access, download, copy, distribute, and use these materials in any way, even commercially, with proper attribution. For these reasons, we cannot publish previously copyrighted maps or satellite images created using proprietary data, such as Google software (Google Maps, Street View, and Earth). For more information, see our copyright guidelines: http://journals.plos.org/plosone/s/licenses-and-copyright.

1.    You may seek permission from the original copyright holder of Figure(s) [2, 4, 7, 8, 9, 10, 11, 12, 13, 14, and 15] to publish the content specifically under the CC BY 4.0 license. 

5. Thank you for stating the following in the Funding Section of your manuscript:

"This work was supported by the Youth Fund of the Gansu University of Chinese Medicine (No. ZQ2015-8), the Natural science foundation of Gansu Provinc（No.1610RJZA066), and China national natural science foundation regional science foundation project (No. 81360615)."

6. Please amend the manuscript submission data (via Edit Submission) to include authors Tiantian Zhu, Tingting Shi, Juan Chen, Ling Jin.

7. Please amend your list of authors on the manuscript to ensure that each author is linked to an affiliation. Authors’ affiliations should reflect the institution where the work was done (if authors moved subsequently, you can also list the new affiliation stating “current affiliation:….” as necessary).

Reviewers' comments:

Reviewer's Responses to Questions

**Comments to the Author**

1. Is the manuscript technically sound, and do the data support the conclusions?

Reviewer #1: Yes

2. Has the statistical analysis been performed appropriately and rigorously? 

Reviewer #1: Yes

3. Have the authors made all data underlying the findings in their manuscript fully available?

Reviewer #1: Yes

4. Is the manuscript presented in an intelligible fashion and written in standard English?

Reviewer #1: Yes

5. Review Comments to the Author

Reviewer #1: The manuscript is good and publishable. Few grammatical errors within the manuscript can be corrected including the addressing the uploaded comments. I recommend it for publication.

Abstract is in order.

Introduction

In lines 57 – 59, authors should explain how expansion of cultivation of A. Sinensis has deteriorated its yield, since it is expected that when something is expanded, it should be increased in quantity. Otherwise, since focus on this research is on ‘quality’, the yield aspect can be omitted.

Materials and Methods

Lines 93 – 94: What is the difference between meteorological data and comprehensive meteorological data used as ecological factors. I think authors could just use one of these and explain further in the methods the kind of meteorological data used in the study.

Results

Presentation of results in lines 214 – 246 is cumbersome and unclear. Authors should simply present the relationships and their significance coherently, which can be presented in a tabular form to show their predictability.

Discussion

Lines 260 – 295: Authors should back their findings with literature to consolidate generated knowledge from this research. Current discussion is more of a commentary on the results other than backing the results with scientific facts on why such findings were obtained and their implications.

Conclusion is in order.

6. PLOS authors have the option to publish the peer review history of their article (what does this mean?). If published, this will include your full peer review and any attached files.

Reviewer #1: No

---

## [Author Response · Author response to Decision Letter 0]

9 Nov 2020

PONE-D-20-27049 Response to Reviewers

18-Nov-2020

Dear Editors and Reviewers:

Thank you for your comments concerning our manuscript entitled “Quality suitability regionalization analysis of Angelica sinensis in Gansu ,China”(ID:PONE-D-20-27049). Those comments are all valuable and very helpful for revising and improving our paper, as well as the important guiding significance to our researches. We have studied comments carefully and have made correction which we hope meet with approval. Revised portion are marked in red in the paper. The corrections in the paper and the responds to the reviewer’s comments are as flowing:

Journal Requirements:

1.Please ensure that your manuscript meets PLOS ONE's style requirements, including those for file naming.

Response: Done. Thank you for your careful review. The manuscript has been corrected as PLOS ONE's style requirements, revised portion were marked in red.

2.Please complete your Competing Interests on the online submission form to state any Competing Interests. If you have no competing interests, please state "The authors have declared that no competing interests exist.", This information should be included in your cover letter; we will change the online submission form on your behalf.

Response: Done. Thank you for your careful review. We have no competing interests. The statement "The authors have declared that no competing interests exist."has been added to the cover letter.

3.Please note that in order to use the direct billing option the corresponding author must be affiliated with the chosen institute. Please either amend your manuscript to change the affiliation or corresponding author, or email us at plosone@plos.org with a request to remove this option.

Response: Done. Thank you for your careful review. The manuscript has been amended,

the corresponding author was affiliated with the chosen institute,now.

4.We note that [Figure(s) 2, 4, 7, 8, 9, 10, 11, 12, 13, 14, and 15] in your submission contain [map/satellite] images which may be copyrighted.We require you to either (1) present written permission from the copyright holder to publish these figures specifically under the CC BY 4.0 license, or (2) remove the figures from your submission.

Response: Done. Thank you for your careful review. Figure(s) 2, 4, 7, 8, 9, 10, 11, 12, 13, 14, and 15 are all the standard maps of Gansu Province, now. Map review number: Gan S(2017)64. Download address: Department of natural resources of Gansu Province (http://zrzy.gansu.gov.cn/).The standard map of Gansu province can be downloaded for free, it does not need to be approved.

5.We note that you have provided funding information that is not currently declared in your Funding Statement.Please remove any funding-related text from the manuscript and let us know how you would like to update your Funding Statement. Please include your amended statements within your cover letter; we will change the online submission form on your behalf.

Response: Done. Thank you for your careful review. The funding-related text from the manuscript has been removed. Funding Statements were added to the cover letter.

6. Please amend the manuscript submission data (via Edit Submission) to include authors Tiantian Zhu, Tingting Shi, Juan Chen, Ling Jin.

Response: Done. Thank you for your careful review. The manuscript submission data has been amended , including authors Tiantian Zhu, Tingting Shi, Juan Chen, Ling Jin.

7. Please amend your list of authors on the manuscript to ensure that each author is linked to an affiliation. 

Response: Done. Thank you for your careful review. The list of authors on the manuscript has been amended,revised portion were marked in red.

8.While revising your submission, please upload your figure files to the Preflight Analysis and Conversion Engine (PACE) digital diagnostic tool,  PACE helps ensure that figures meet PLOS requirements.

Response: Done. Thank you for your careful review. The figure files have been uploaded to PACE. All the figures meet PLOS requirements, now.

Reviewer(s)' Comments to Author:

1.In lines 57 – 59, authors should explain how expansion of cultivation of A. Sinensis has deteriorated its yield, since it is expected that when something is expanded, it should be increased in quantity. Otherwise, since focus on this research is on ‘quality’, the yield aspect can be omitted.

Response: Done. Thank you for your careful review.We are so sorry about the wrong statement.The sentence has been corrected as“… which eventually has deteriorated the quality of A. sinensis”, Please see the red part in Line 57-58 of Page 3.

2.Materials and Methods

Lines 93 – 94: What is the difference between meteorological data and comprehensive meteorological data used as ecological factors. I think authors could just use one of these and explain further in the methods the kind of meteorological data used in the study.

Response: Done. Thank you for your careful review.We are so sorry about the wrong statement.We've deleted “comprehensive meteorological data”.Please see the red part in Line 91-92 of Page 5.

3.Results

Presentation of results in lines 214 – 246 is cumbersome and unclear. Authors should simply present the relationships and their significance coherently, which can be presented in a tabular form to show their predictability.

Response: Done. Thank you for your careful review. Presentation of results in lines 214 – 246 have been presented in a tabular form to show their predictability.Please see the red part in Line 211-218 of Page12.

4.Discussion

Lines 260 – 295: Authors should back their findings with literature to consolidate generated knowledge from this research. Current discussion is more of a commentary on the results other than backing the results with scientific facts on why such findings were obtained and their implications.

Response: Done. Thank you for your careful review. Literature has been added to support our findings to consolidate the knowledge derived from this study. Please see the red part in Line 239-267 of Page15.

Special thanks to Editors and Reviewers for these good comments.

We appreciate for Editors and Reviewers’ warm work earnestly, and hope that the correction will meet with approval.

Once again, thank you very much for your comments and suggestions.

---

## [Editor Report · Decision Letter 1]

26 Nov 2020

Quality suitability regionalization analysis of Angelica sinensis in Gansu , China

PONE-D-20-27049R1

Dear Dr. Jin,

We’re pleased to inform you that your manuscript has been judged scientifically suitable for publication and will be formally accepted for publication once it meets all outstanding technical requirements.

Kind regards,

Vassilis G. Aschonitis

Academic Editor

PLOS ONE
---

## [Editor Report · Acceptance letter]

3 Dec 2020

PONE-D-20-27049R1 

Quality suitability regionalization analysis of *Angelica sinensis* in Gansu, China 

Dear Dr. Jin:

I'm pleased to inform you that your manuscript has been deemed suitable for publication in PLOS ONE. Congratulations! Your manuscript is now with our production department. 

Kind regards, 

on behalf of

Dr. Vassilis G. Aschonitis 

Academic Editor

PLOS ONE